# Relationship between Muscle-Tendon Stiffness and Drop Jump Performance in Young Male Basketball Players during Developmental Stages

**DOI:** 10.3390/ijerph192417017

**Published:** 2022-12-18

**Authors:** Marco Gervasi, Piero Benelli, Roberto Venerandi, Eneko Fernández-Peña

**Affiliations:** 1Department of Biomolecular Sciences, Division of Exercise and Health Sciences, University of Urbino Carlo Bo, 61029 Urbino, Italy; 2US Victoria Libertas Basketball, 61122 Pesaro, Italy; 3Department of Physical Education and Sport, University of the Basque Country UPV/EHU, 01007 Vitoria-Gasteiz, Spain

**Keywords:** biomechanics, team sports, reactive strength index, patellar tendon, gastrocnemius-Achilles tendon unit, quadriceps tendon, rectus femoris, drop jump

## Abstract

Background: The relationship between stiffness and drop jump performance in athletes in various stages of development has yet to be fully investigated. The first aim of this study was to investigate the association between the stiffness of the patellar and quadriceps tendon (PT, QT), gastrocnemius–Achilles tendon unit (GAT), and rectus femoris (RF) using drop jump (DJ) performance in young basketball players. The second aim was to investigate possible variations in the stiffness levels of those tissues in different developmental stages. Methods: The stiffness levels of the GAT, PT, QT, and RF were measured in both limbs in 73 male basketball players aged 12 to 18 years. The reactive strength index (RSI), contact time (CT) and jump height (JH) during 30 and 40 cm DJs were also measured. Results: Pearson correlation coefficients showed a significant association between DJ performance and PT, QT, GAT, and RF dynamic stiffness. Moreover, the youngest subjects were found to have lower stiffness values than the older ones. Conclusions: Tissue stiffness can affect athletic performance by modifying the stretch-shortening cycle in young basketball players. Stiffness of muscles and tendons increases during the maturation process. Further investigations could shed light on the effect of training on the stiffness of muscles and tendons.

## 1. Introduction

Currently, more than 450 million people in the world play basketball, making it among the five most popular sports on the planet [1]. Jumping ability has been shown to be a predictor of functional capacity [2,3,4] and the risk of falling [5], as well as a strong predictor of basketball performance both in elite and recreational players [6,7]. When humans jump, the tendon structures are assumed to be the major source of the elastic component; hence, the elasticity of tendon structures is a leading factor contributing to the amount of stored energy [8]. In sports with stretch-shortening cycle (SSC) movements, such as jumps in basketball, a stiffer muscle-tendon unit may be required for the storage and release of elastic energy [8,9]. Given the young age at which athletes begin playing basketball, this structure undergoes a development process that runs through puberty and probably contributes to the performance of high-level athletes. This is one of the reasons why the most common overuse injuries in this sport involve the knees (patellofemoral pain or “jumper’s knee”) and ankles (Achilles tendon pain) [10,11,12]. In addition, athletic training often aims to achieve higher levels of lower limb stiffness, as it is associated with better repeated sprint ability and horizontal jumping performance in basketball players [13,14,15].

Among the most commonly used tests to assess vertical jump capacity, the drop jump is linked to the fastest stretch-shortening cycle [16], since the floor contact duration in adults is under 250 ms [17]. Therefore, having stiffer tendons and/or muscles might be beneficial for a more effective force transmission from the contractile element via the tendon to the bones [18]. However, although a strong relationship between stiffness and performance indices in vertical jumping is hypothesized, there is limited and contrasting evidence to support this relationship [19,20,21]. This is likely due to the different methods and protocols that have been used to measure stiffness and jump performance, as well as the different training levels and backgrounds of the subjects examined. For example, several authors reported moderate positive correlations of gastrocnemius medialis muscle stiffness (measured by ultrasound shear wave elastography or SWUE) with jump height and the reactive strength index (RSI = jump height/contact time) in healthy men during a 15 cm drop jump, but not with contact time [21]. In contrast, other authors did not find any positive correlations between 40 cm drop jump performance parameters and quadriceps and triceps surae stiffness (assessed with the MyotonPRO device both at the tendons and muscle tissue) in recreationally active adult males [19].

Previous studies have shown that tendon stiffness increases during maturation, mainly because of muscle growth and a higher mechanical load due to larger body mass [22]. Mersmann et al. (2017) [23], for instance, showed that patellar tendon stiffness of the jumping leg increased from mid-to late adolescence (16 to 18 years) in male and female volleyball players. They proposed the enlargement of the tendon cross sectional area as the main contributor to higher stiffness and suggested that this adaptation has the potential to improve athletic performance.

However, to date, there is still a lack of studies investigating muscle and tendon stiffness in a wider age range, as well as their direct relationship with sporting performance during the maturation process. Specifically, young players in team sports in which jumping abilities are very important could be a good model to study the relationship between jump mechanical load and muscle and tendon tissue adaptations from early to late adolescence [13,24].

In recent years, technological developments have made it possible to create instruments capable of accurately and non-invasively measuring and analyzing the viscoelastometric parameters of tendons and muscles in subjects with pathologies, healthy subjects and athletic subjects [25,26,27]. Recently [28,29], SWUE has been shown to be a valid and reliable tool for the quantification of the elastic properties of healthy [29,30] and pathological [31] tendons, but it has high initial and maintenance costs. Fortunately, a lower-cost alternative called MyotonPRO (MyotonPRO, Myoton Ltd., Estonia) has proven to be a reliable, accurate, and sensitive tool that can be used for the objective non-invasive digital palpation of superficial skeletal muscles [32] and tendons [33]. It has recently been shown that MyotonPRO is related to the Young’s modulus of muscles and tendons (gastrocnemius and Achilles tendons) quantified by SWUE, and thanks to its good intra-operator repeatability, it can be used for the assessment of the mechanical properties (such as stiffness) of the muscle belly and tendons in resting conditions [34].

The relationship between MyotonPRO measurements and drop jump performance in athletes during various developmental stages has yet to be investigated. Therefore, the first aim of the present transversal study was to investigate potential relationships between lower limb tendon and muscle stiffness and the performance parameters of 30- and 40-cm drop jumps (DJ30 and DJ40). The second aim was to investigate possible variations in the stiffness levels of these tissues in different developmental stages of the maturation process in basketball players aged 12 to 18 years.

## 2. Materials and Methods

### 2.1. Participants

The a priori power analysis was performed using G*Power software (version 3.1.9.4) considering a two way ANOVA test: with an effect size f = 0.4, α = 0.05 and statistical power (1−β) = 0.85, a total of 72 subjects must be enrolled. Given this outcome, seventy-three healthy young males, aged 12 to 18 years, took part in the study. In order to analyze variations in stiffness across developmental stages, the subjects were divided into three age groups (≤13, 14–15 and ≥16 years). To identify the age at peak height velocity (APHV), the maturity offset (MO) of the three groups was estimated using the alternative model for boys by Moore et al. (2015) [35]. The subjects’ age, weight, height, BMI and MO are displayed in Table 1. All of the participants started playing mini basketball on a feeder team of an Italian first division team at the age of 6 to 8 years. At the time of the data collection, they were practicing basketball 3 to 5 times per week for an average of 7.2 h per week. The participants had no history of surgery or any other orthopedic injuries in the previous year and were not following any particular therapy. All participants signed an informed consent form, and in the case of underage subjects, the consent form was signed by their parents. The study was approved by the Human Research Ethical Committee of the Urbino University (No. 31_2020) and was performed in accordance with the guidelines of the Declaration of Helsinki. All tests were performed in a controlled setting (in the same gym, with the same operators, and with the same instruments) during the basketball season.

### 2.2. Procedures

The assessment of stiffness was performed after a warm-up consisting of 5 min of pedaling at moderate intensity, followed by 5 min of squat jump exercises and dynamic stretching. The dynamic stiffness of the gastrocnemius–Achilles tendon unit (GAT), the Patellar Tendon (PT), the Quadriceps Tendon (QT) and the Rectus Femoris (RF) was measured on the left and right sides using the MyotonPro device. For GAT assessment, participants were in a prone position on a massage table with their knees extended and feet hanging barefoot and unsupported from the edge of the table. The measurement points were at 8, 9, 10, 11, 12, 16, and 20 cm from the plantar aspect of the heel, as shown in Figure 1 [33]. For the PT assessment, participants were seated with their knees flexed at a 90° angle and completely relaxed. The measurement point was at the midpoint between the distal edge of the patella and the tibial tuberosity. For the RF and QT assessments, participants were lying supine with their feet on the massage table. The RF was assessed at the midpoint between the anterior superior iliac spine and the superior edge of the patella, whereas the QT was assessed at the distal 1/3 point between the RF assessment point and the upper edge of the patella.

Drop jump performance was assessed immediately after the stiffness measurements were taken using an optoelectronic system called Optojump (Microgate s.r.l., Bolzano, Italy). This device consists of a transmitting and receiving bar, each having 96 LEDs. Continuous communication takes place between the LEDs on the transmitting and receiving bars, and the mechanism determines the duration of any communication interruptions between them. This allows for 1/1000-s precise measurement of flight and contact times throughout the execution of a sequence of jumps. Participants performed three DJ30s and three DJ40s with their hands placed on their hips, with 30 s of passive rest between jumps. The subjects were instructed to keep their leg muscles as stiff as possible, trying to minimize joint flexions during landing. The main goal of the DJ was to minimize landing contact time while trying to reach the maximum jump height. Data on contact time (CT30 and CT40), jump height (JH30 and JH40) and the RSI (RSI30 and RSI40) of each drop jump were then calculated and the best DJ30 and DJ40 (defined as the highest jump) were selected for subsequent analysis. Specifically, the RSI is an index of reactive strength derived from the ratio of jump height to ground contact time. It makes it possible to assess the subject’s ability to change rapidly from eccentric to concentric muscle contraction as well as his or her explosiveness during the drop jump [36].

### 2.3. Statistical Analysis

All data were checked for normality using the Shapiro–Wilk test and for equality of variances using Levene’s test. A Pearson’s correlation (rP) analysis was then used to calculate the relationship between drop jump performance variables and muscle or tendon stiffness values. The alpha level was initially set to 0.05, but was then corrected to 0.0236 using the Benjamini-Hochberg procedure to decrease the false discovery rate (FDR < 5%). The effect sizes of rP were established as described by Hopkins and defined as trivial (r = 0–0.1), small (r = 0.1–0.3), moderate (r = 0.3–0.5), large (r = 0.5–0.7), very large (r = 0.7–0.9) and nearly perfect (r = 0.9–1.0) [37]. GAT data were analyzed by means of a two-way ANOVA for independent measures (age and measurement point as independent factors). Jump performance parameters, PT, RF and QT data were analyzed using a one-way ANOVA with age as an independent factor. To identify which groups differed from each other, a post-hoc analysis was performed using the Bonferroni procedure. All data were analyzed using Microsoft Excel 2016 and JASP (Version 0.16.4) (University of Amsterdam, Amsterdam, The Netherlands).

## 3. Results

### 3.1. Drop Jump Performance Parameters in Different Age Groups

Drop jump performance parameters of all participants and age groups are shown in Table 2. There were differences in RSI and JH data between the ≥16 age group and the younger groups, but no differences were detected between ≤13 and 14–15 age groups for any variable. CT remained constant among all age groups.

### 3.2. Correlation between Drop Jump Performance Parameters and Stiffness

When considering the whole group of participants, a Pearson’s correlation analysis showed that RSI and JH were related to the stiffness of PT, RF and GAT at 8, 9, 10 and 12 cm in both the right and left limbs. In particular, DJ40 showed a larger effect on performance parameters than DJ30, as shown in Table 3 and Table 4. Figure 2 depicts the correlations between RSI40 and JH40 with RF and PT stiffness. On the other hand, no effects or negligible effects were detected for contact time, GAT at 16 and 20 cm, while QT stiffness showed a significant correlation with JH only in the left limb at both DJ30 and DJ40.

### 3.3. Tendon and Muscle Stiffness in Different Age Groups

The analysis of GAT stiffness among the age groups showed a significant increase from 12 to 18 years for both left and right limbs, with all age groups found to be significantly different from each other (*p* < 0.001), as shown in Figure 3. A post hoc analysis showed that GAT stiffness decreased from 8 to 20 cm in all age groups, with significant differences among all measurement points when the distance was 2 cm or larger (*p* < 0.001), except for 8–10 cm (*p* = 0.012) and 9–11 cm (*p* = 0.067).

Similarly, for both left and right limbs, RF, PT and QT stiffness showed a significant increase from 12 to 18 years (Figure 4). In particular, for RF, the muscle stiffness of the ≤13 group was lower than that found in the 14–15 (p < 0.001) and ≥16 groups (*p* < 0.001), while no difference was found between the 14–15 and ≥16 age groups (*p* = 0.069). For PT, the tendon stiffness of the ≤13 group was lower than that of the 14–15 (*p* = 0.028) and ≥16 groups (*p* = 0.018), while no difference was found between the 14–15 and ≥16 age groups (*p* = 1.000). The same result was found for QT, with the tendon stiffness of the ≤13 group found to be lower than that of the 14–15 (*p* = 0.009) and ≥16 groups (*p* = 0.002), while no difference was found between the 14–15 and ≥16 age groups (*p* = 1.000).

No differences were found between the left and right limbs for any stiffness parameter or GAT measurement points.

## 4. Discussion

### 4.1. Pearson’s Correlation Analysis between Stiffness and DJ Performance Parameters

The first aim of this study was to investigate the potential relationships between lower extremity tendon and muscle stiffness with the performance parameters of DJ30 and DJ40. First, as expected, jump height and RSI were found to improve from ages 12 to 18 years, while contact time did not change for DJ30 and DJ40, except for a decreasing trend in the ≥16 age group (see Table 2). After a Pearson’s correlation analysis, we found that the stiffness of GAT (except for 16 and 20 cm), RF, PT and QT showed a small to moderate correlation with RSI and JH (Figure 2), and the effect size of all correlations increased with box height (DJ30 to DJ40). Nevertheless, we found no correlation between stiffness and CT in young basketball players. These findings suggest that drop jump performance may be influenced by the GAT, RF, PT and QT stiffness at rest.

These results are in line with a recent investigation by Ando, et al. (2021) [21], which showed that medial gastrocnemius stiffness (measured during a 15 cm drop jump as the shear modulus through ultrasound shear wave elastography) correlated with the same performance parameters (RSI and JH) but not with the contact time. In addition, our results are in line with other studies [38,39] in which the authors demonstrated that drop jumps performed with different modes and instructions show different responses. In particular, the authors showed that when participants were asked to reach the maximum jump height or to make their best effort, without further instruction, they obtained the best JH, with no correlation with contact time. On the other hand, when participants were asked to keep their limbs as stiff as possible, they obtained lower JH and there was a significant correlation with contact time. In both cases, however, the authors found a strong correlation with RSI. Furthermore, the authors found that the jumps with the greatest height were strongly correlated with ground reaction force at the start of the propulsive phase. Likewise, Walsh et al. (2004) [40] also reported that the greatest power output occurred in moderate stiffness drop jumps. Therefore, it is not surprising that contact time was not related to jump height, as the time required to transfer high reaction forces varies with the athlete’s rate of force development capacity. In our study, the instructions given to the participants were to maintain the minimum contact time by seeking the greatest possible height, and for data analysis, we collected the highest DJ30 and DJ40. Indeed, this may be why we found no correlation between stiffness levels and contact time.

Regarding the RF, PT and QT stiffness, this is the first study to investigate and find a significant correlation with RSI and JH (Figure 2). Not surprisingly, the rectus femoris muscle, quadriceps and patellar tendon were shown to be directly involved in the execution of the drop jump, in particular during the landing phase [41,42]. The relationship between RF, PT and QT stiffness and DJ performance parameters is linked with risk of injury, in particular patellar tendinopathies. In fact, it is known that the ability of the patellar and quadriceps tendons to lengthen during the jump can be compromised by tendinopathy, which reduces the range of movement during DJ landings [43]. Many studies [44,45,46] state that smaller flexion ranges of the lower body joints (stiff landing), larger moment/torque at the knee, and higher knee angular velocities are risk factors for patellar tendinopathy, as well as indicators of present or past quadriceps or patellar injury. Hence, the positive relationship shown in the present study between QT and PT stiffness (assessed with MyotonPro) and DJ performance could be useful in further investigations for the prevention of jumper’s knee. It is worth noting that, surprisingly, for the QT stiffness, we only found a correlation for the left limb with DJ30 and DJ40. We speculate that the different ways of measuring QT with the gold standards SWUE (30° flexed knee) and MyotonPro (extended knee) [47] may have reduced the validity of this measurement. Therefore, we believe that specific studies are needed to define the correct knee flexion angle or the measurement site before continuing to assess QT stiffness by means of myotonometry.

Finally, it is worth mentioning the study by Konrad et al. (2022) [19], who found no correlation between Achilles tendon and patellar stiffness and DJ40 performance parameters (RSI, JH, and CT) using the MyotonPro. One possible explanation for this discrepancy could be (in addition to the smaller number of subjects) that their participants were physically active adults lacking experience in jumping skills, whereas our participants were young basketball players who were familiar with the DJ and good jumping technique. Moreover, it is worth noting that the extensive training experience (about 5 to 10 years) of our participants could have affected their stiffness levels, but we cannot discriminate between the effects of training and maturation alone because of the lack of an untrained control group.

### 4.2. Stiffness of GAT, QT, PT and RF through Developmental Stages

The second aim of this study was to examine how tissue stiffness changes through various developmental stages. Our results show that, in splitting the participants into three age groups, all GAT stiffness values increase from ≤13 to ≥16 years; also, as expected and also shown by Morgan et al. (2018) [48], in all groups the highest values were from 8 to 12 cm, after which they decrease to 20 cm and the differences become smaller with age (see Figure 3). These results suggest that the stiffest component of the Achilles tendon is in its distal portion, and that this portion grows significantly from 12 to 18 years in young basketball players. Mean values of RF, PT and QT also increase significantly from ≤13 to ≥16 years, but unlike GAT, they slow down from 14–15 to ≥16 years. However, unlike tendon stiffness, RF values also increased quasi significantly (*p* = 0.069) from 14–15 to ≥16 years, indicating that the growth of muscle stiffness probably continues more consistently than PT and QT over the developmental stages examined in the present study (see Figure 4). In accordance with our results, several studies have reported increases in muscle cross sectional area (CSA) during maturation, with some suggesting that the greatest changes occur in boys around the age of 13–15 years [49,50]. A roughly two -fold rise in patellar tendon CSA from childhood to adulthood has also been noted in earlier research [51], which suggests an increase in tendon stiffness given the relationship between tendon CSA and stiffness [52]. Both patellar [22,51] and Achilles tendon stiffness [53] have also shown approximately two-fold increases in magnitude with age. Additionally, these studies found no or very little difference between older children (>14 years) and adults, supporting the notion that adult values can be attained shortly after the time of peak height velocity [51,54,55].

A limitation of this study is that MO was assessed using the alternative model for boys by Moore et al. (2015) [35], which uses height instead of sitting height. However, our results show that the three age groups matched the biological and chronological ages, providing an overall picture of the maturation process. As a practical application, coaches and practitioners could use the JH and RSI of a drop jump to roughly monitor the increase of stiffness level during the maturation process and accordingly adjust the training load. A direct measurement of stiffness is still advised, especially during the sensitive phases of growth such as APHV.

## 5. Conclusions

This study showed for the first time that muscle and tendon stiffness measured with MyotonPro is related to drop jump performance in young male basketball players. The stiffness of GAT, RF, PT and QT increase with age during adolescence, but the effects of training are still unclear. Further investigation is warranted to clarify the effects of training on muscle and tendon stiffness and the implications of stiffness for injury risk and incidence in young athletes.

## Figures and Tables

**Figure 1 ijerph-19-17017-f001:**
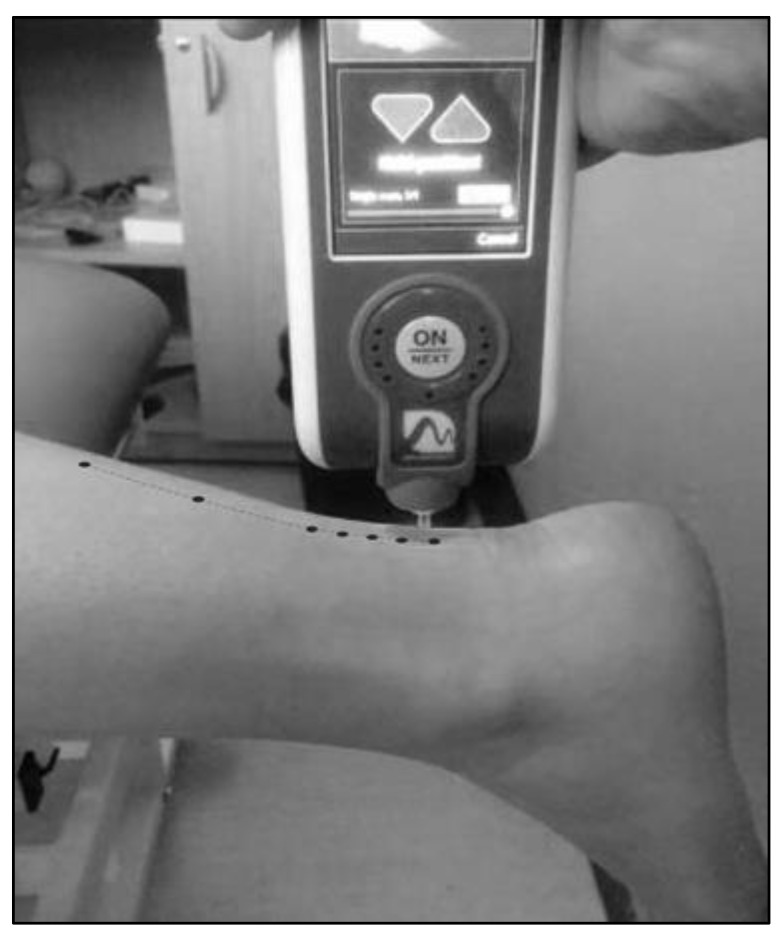
Gastrocnemius–Achilles tendon unit (GAT) stiffness measurement at 8, 9, 10, 11, 12, 16, and 20 cm from the plantar aspect of the heel. Note that the participant is in a prone position with the knees extended and feet hanging barefoot and unsupported from the edge of the table.

**Figure 2 ijerph-19-17017-f002:**
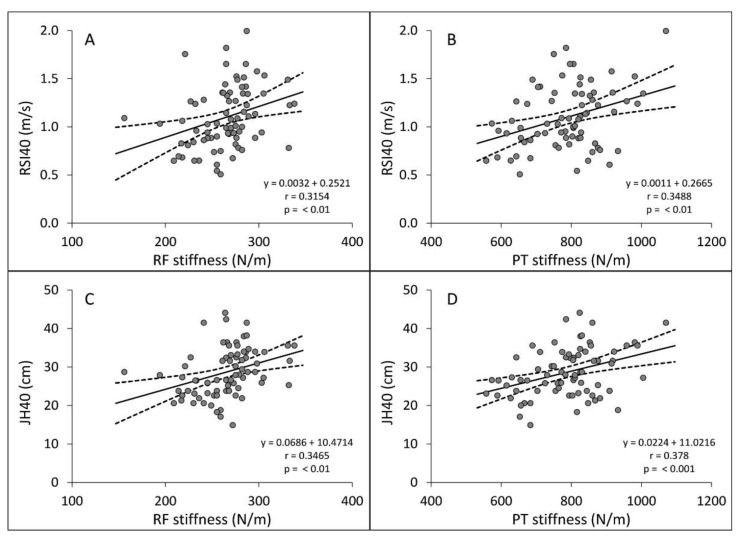
Scatterplots illustrating the significant correlations between Jump Height (JH40) and Reactive Strength Index (RSI40) during the drop jump at 40 cm, with the stiffness of Rectus Femoris (RF) and Patellar Tendon (PT): (**A**) RF stiffness and RSI40; (**B**) PT stiffness and RSI40; (**C**) RF stiffness and JH40; and (**D**) PT stiffness and JH40. The solid lines represent the regression lines. The 95% confidence interval of the correlations is represented by dashed lines. The regression equations are shown in the bottom-right angle of each scatterplot, along with the r value and significance level of p.

**Figure 3 ijerph-19-17017-f003:**
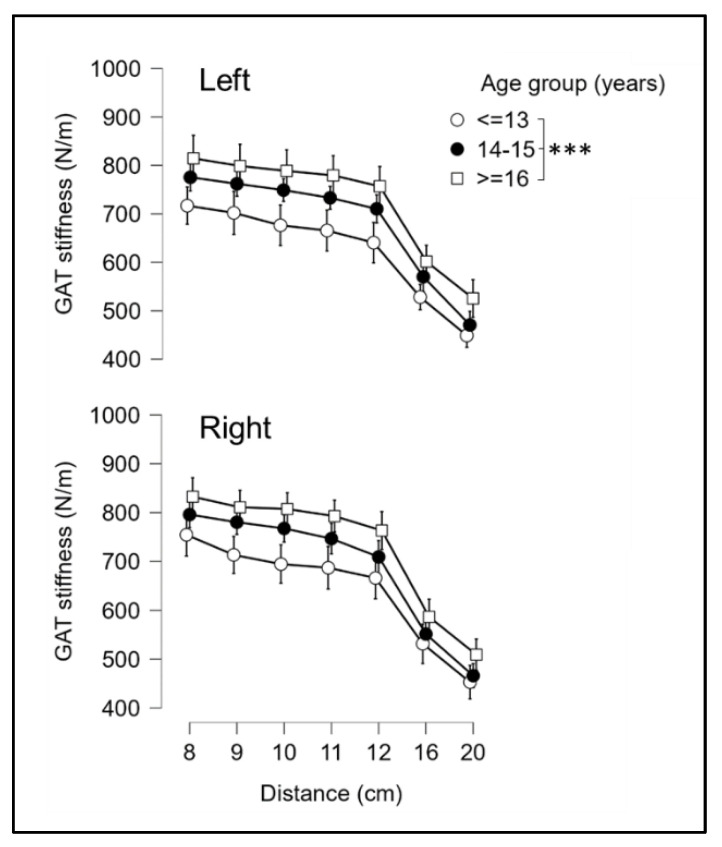
Gastrocnemius–Achilles tendon unit (GAT) stiffness at 8, 9, 10, 11, 12, 16, and 20 cm from the plantar aspect of the heel for the three age groups in left (**left**) and right (**right**) limbs. Error bars represent a confidence interval of 95%. *** = significantly different between groups (*p* < 0.001).

**Figure 4 ijerph-19-17017-f004:**
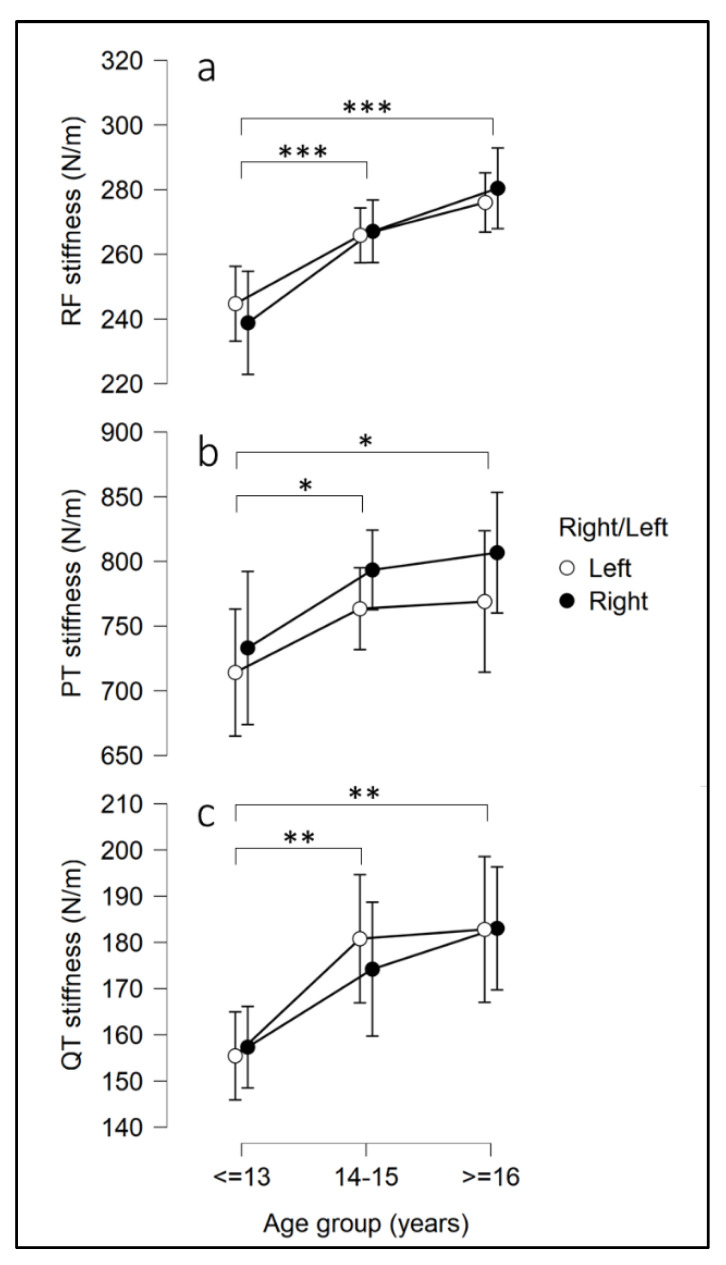
Rectus Femoris muscle stiffness (**a**, RF), Patellar Tendon stiffness (**b**, PT) and Quadriceps Tendon stiffness (**c**, QT) in the three age groups in left and right limbs. Error bars represent a confidence interval of 95%. * = *p* < 0.05; ** = *p* < 0.01; *** = *p* < 0.001.

**Table 1 ijerph-19-17017-t001:** Demographic and anthropometric parameters of the participants (mean ± SD).

Parameter	All Participants (*n* = 73)	≤13 Years (*n* = 18)	14–15 Years (*n* = 34)	≥16 Years (*n* = 21)
Age (years)	15.2 ± 1.7	13.2 ± 0.4	15.0 ± 0.6	17.3 ± 1.0
Weight (kg)	67.9 ± 14.2	53.9 ± 9.0	67.4 ± 10.2	80.8 ± 11.4
Height (m)	1.79 ± 0.1	1.68 ± 0.1	1.79 ± 0.1	1.88 ± 0.1
BMI (kg/m^2^)	21.0 ± 2.7	19.0 ± 2.3	20.9 ± 2.2	22.9 ± 2.4
MO (years)	1.9 ± 1.5	0.0 ± 0.5	1.7 ± 0.7	3.7 ± 0.9

BMI = Body Mass Index; MO = Maturity Offset.

**Table 2 ijerph-19-17017-t002:** Drop jump performance parameters of the participants for DJ30 and DJ40 (mean ± SD).

Parameter	All Participants (*n* = 73)	≤13 Years (*n* = 18)	14–15 Years (*n* = 34)	≥16 Years (*n* = 21)
RSI30 (m/s)	1.02 ± 0.26	0.89 ± 0.23 ***	0.97 ± 0.24 **	1.19 ± 0.24
JH30 (m)	0.27 ± 0.06	0.23 ± 0.05 ***	0.27 ± 0.05 **	0.32 ± 0.05
CT30 (s)	0.275 ± 0.05	0.265 ± 0.05	0.282 ± 0.05	0.273 ± 0.05
RSI40 (m/s)	1.09 ± 0.31	0.92 ± 0.35 ***	1.07 ± 0.32	1.26 ± 0.26
JH40 (m)	0.28 ± 0.06	0.24 ± 0.05 ***	0.28 ± 0.06 **	0.33 ± 0.06
CT40 (s)	0.270 ± 0.05	0.274 ± 0.04	0.271 ± 0.05	0.267 ± 0.04

RSI = Reactive Strength Index (JH/CT); JH = Jump Height; CT = Contact Time. ** = significantly different from ≥16 years group (*p* < 0.01). *** = significantly different from ≥16 years group (*p* < 0.001). No statistical differences were found between ≤13 and 14–15 age groups.

**Table 3 ijerph-19-17017-t003:** Correlations between DJ30 performance parameters and stiffness.

Parameter		Tendon and Muscle Stiffness									
		GAT							RF	PT	QT
		8 cm	9 cm	10 cm	11 cm	12 cm	16 cm	20 cm			
Right RSI30	r	0.21	0.31	0.27	0.24	0.20	0.12	0.12	0.38	0.27	0.18
	*p*	0.07	**0.01**	**0.02**	0.04	0.09	0.29	0.29	**0.00**	**0.02**	0.12
Left RSI30	r	0.31	0.24	0.27	0.29	0.27	0.21	0.14	0.37	0.24	0.18
	*p*	**0.01**	0.04	**0.02**	**0.01**	**0.02**	0.07	0.22	**0.00**	0.03	0.11
Right JH30	r	0.18	0.33	0.31	0.29	0.21	0.12	0.14	0.32	0.20	0.20
	*p*	0.13	**0.00**	**0.01**	**0.01**	0.07	0.31	0.23	**0.01**	0.08	0.08
Left JH30	r	0.26	0.20	0.28	0.28	0.28	0.13	0.17	0.27	0.17	0.27
	*p*	**0.02**	0.08	**0.02**	**0.01**	**0.02**	0.25	0.15	**0.02**	0.15	**0.02**
Right CT30	r	−0.08	−0.05	−0.02	0.01	−0.01	−0.02	−0.03	−0.15	−0.15	−0.01
	*p*	0.52	0.66	0.88	0.91	0.91	0.86	0.78	0.19	0.19	0.94
Left CT30	r	−0.12	−0.10	−0.06	−0.07	−0.06	−0.12	0.00	−0.19	−0.13	0.05
	*p*	0.29	0.38	0.63	0.53	0.62	0.29	0.98	0.11	0.25	0.68

RSI = Reactive Strength Index (JH/CT); JH = Jump Height; CT = Contact Time. GAT = Gastrocnemius–Achilles Tendon unit; PT = Patellar Tendon; QT = Quadriceps Tendon; RF = Rectus Femoris muscle. r = Pearson’s correlation coefficient. *p* = *p* value. The alpha level was set to 0.0236 after the Benjamini-Hochberg procedure. Significant values are in bold.

**Table 4 ijerph-19-17017-t004:** Correlations between DJ40 performance and stiffness parameters (mean ± SD).

Parameter		Tendon and Muscle Stiffness									
		GAT							RF	PT	QT
		8 cm	9 cm	10 cm	11 cm	12 cm	16 cm	20 cm			
Right RSI40	r	0.25	0.37	0.33	0.32	0.27	0.21	0.15	0.32	0.35	0.10
	*p*	0.03	**0.00**	**0.00**	**0.01**	**0.02**	0.07	0.19	**0.01**	**0.00**	0.40
Left RSI40	r	0.37	0.32	0.37	0.35	0.35	0.25	0.19	0.31	0.30	0.12
	*p*	**0.00**	**0.01**	**0.00**	**0.00**	**0.00**	0.03	0.10	**0.01**	**0.01**	0.30
Right JH40	r	0.23	0.33	0.33	0.30	0.24	0.23	0.24	0.35	0.38	0.24
	*p*	0.05	**0.00**	**0.00**	**0.01**	0.03	0.05	0.03	**0.00**	**0.00**	0.04
Left JH40	r	0.31	0.27	0.34	0.34	0.33	0.20	0.27	0.32	0.32	0.32
	*p*	**0.01**	**0.02**	**0.00**	**0.00**	**0.00**	0.09	0.03	**0.00**	**0.00**	**0.01**
Right CT40	r	−0.12	−0.21	−0.13	−0.14	−0.12	−0.03	0.02	−0.12	−0.11	0.09
	*p*	0.32	0.07	0.27	0.23	0.32	0.79	0.83	0.32	0.37	0.43
Left CT40	r	−0.20	−0.18	−0.18	−0.15	−0.14	−0.14	0.02	−0.12	−0.06	0.16
	*p*	0.08	0.12	0.13	0.19	0.23	0.22	0.85	0.30	0.62	0.17

RSI = Reactive Strength Index (JH/CT); JH = Jump Height; CT = Contact Time. GAT = Gastrocnemius–Achilles Tendon unit; PT = Patellar Tendon; QT = Quadriceps Tendon; RF = Rectus Femoris muscle. r = Pearson’s correlation coefficient. *p* = *p* value. The alpha level was set to 0.0236 after the Benjamini-Hochberg procedure. Significant values are in bold.

## Data Availability

Not applicable.

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
