# Peer review of "Relationship between Muscle-Tendon Stiffness and Drop Jump Performance in Young Male Basketball Players during Developmental Stages"

_ijerph, 2022, doi:10.3390/ijerph192417017_

Round 1
Reviewer 1 Report
GENERAL COMMENTS
With the present transversal study, the authors aimed to investigate potential relationships between lower limb tendon and muscle stiffness and the performance parameters of 30- and 40-cm drop jumps. Also, they aimed to investigate possible variations in the stiffness levels of these tissues in different developmental stages of the maturation process in basketball players aged 12 to 18 years.
This paper is well-written, and the topic is interesting.
I have only a few comments for the authors to improve the manuscript.
SPECIFIC COMMENTS
Abstract
It is written correctly. Gives highlights from each section of the paper.
Introduction
The authors did a good job of synthesizing the literature.
The reasoning logically follows the pattern correctly: Known → Unknown → Research question/hypothesis.
The gaps in the literature to be filled are described.
Methods
Overall, the methodology is clearly explained.
The tools used are validated and reliable.
The statistical techniques used are appropriate.
Participants
-How did you decide on the sample size before starting the study? Have you carried out an a priori statistical power analysis? Please describe the alpha error, statistical power, and effect size used to determine the sample size a priori.
Results
The results are written correctly.
The figures and tables are explanatory.
Discussion
The discussions are clear and to the point.
The limitations are described.
Conclusions
The authors' conclusions are justified. The take-home message is clear.
Reviewer 2 Report
The aim of the present study is to investigate relationships between lower limb tendon and muscle stiffness and the drop jumps, and the effect of maturation status on these relationships. The work has very well-defined objectives and the sample size and methodology seem adequate to try to meet the objectives.
However, in my opinion, there are several aspects that need to be improved before publication:
The authors point out that a possible limitation of the study is that the maturational state was not measured. Knowing the sex, height, and age it is possible to estimate the maturation stage. (Moore, S. A., McKay, H. A., Macdonald, H., Nettlefold, L., Baxter-Jones, A. D., Cameron, N., & Brasher, P. M. (2015). Enhancing a Somatic Maturity Prediction Model. Medicine and science in sports and exercise, 47(8), 1755–1764. https://doi.org/10.1249/MSS.0000000000000588 and/or the other possible equations)
In the methods section it would be interesting to include some type of flow diagram or images of the tests so that people with little knowledge of this measurements could understand the method well (this work is not only interesting for coaches / scientists)
The presentation of the results should be improved by using more correlation matrices and summarizing the information in the text. It would be interesting to include one or two scatterplots with the strongest or most interesting correlations.
Include a paragraph with the practical applications of the work at the end of the discussion section.
Please revise the reference section according to the journal rules.
